# A multi-disciplinary approach to identify spillover interfaces of bat coronaviruses to pig farms in Italy

**Francesca Festa**[1,2]*, **Pamela Priori**[3], **Giulia Chiarello**[4¤a], **Elisa Palumbo**[1], **Gianpiero Zamperin**[1], **Francesca Cosentino**[2¤b], **Luigi Maiorano**[2], **Maria Luisa Menandro**[4], **Dino Scaravelli**[3], **Paola De Benedictis**[1], **Pierre Nouvellet**[5,6], **Stefania Leopardi**[1,7]

**1** Department of Virology, Research and Innovation Health, Istituto Zooprofilattico Sperimentale delle Venezie, Padua (PD), Italy, **2** Department of Biology and Biotechnology "Charles Darwin", Sapienza University of Rome (RM), Rome, Italy, **3** Cooperativa S.T.E.R.N.A., Ecological STudies Research Nature Environment, Forlì (FC), Italy, **4** Department of Animal Medicine, Production and Health, University of Padua, Padua (PD), Italy, **5** Department of Evolution, Behaviour, and Environment, School of Life Sciences, University of Sussex Brighton, Brighton and Hove, United Kingdom, **6** Department of Infectious Disease Epidemiology, School of Public Health, Imperial College London, London, United Kingdom, **7** Department of Veterinary Medicine, Università Aldo Moro di Bari, Valenzano, Italy

¤a Current address: Laboratory of Medical Entomology and Vector-borne Diseases, Istituto Zooprofilattico Sperimentale delle Venezie, Padua (PD), Italy
¤b Current address: Department of Biotechnology and Biosciences, University of Milano-Bicocca, Milano (MI), Italy
* ffesta@izsvenezie.it

## Abstract

Bats are recognized reservoirs of diverse coronaviruses (CoVs), but little is known about the pathways enabling their spillover into livestock. This study applied a multi-disciplinary approach, combining bioacoustic surveys, landscape analysis and molecular virology, to assess the risk of CoV transmission from bats to pigs in intensive farming systems of Northern Italy. Between 2021 and 2022, we carried out bioacoustic monitoring in 14 pig farms to assess bat presence, diversity and behaviour. We also analysed landscape and farm-level variables to identify predictors of bat activity and richness. Additionally, we investigated CoV circulation in three populations of *Pipistrellus kuhlii* through active longitudinal surveillance, performing whole-genome sequencing on new and archival CoV strains detected in *P. kuhlii* and *Hypsugo savii*. Using these data, we explored the viral biodiversity potentially present at this interface via genetic and phylogenetic analyses. We identified eight bat species across farms, with *P. kuhlii*, *P. pipistrellus* and *H. savii* being the most widespread and active. Landscape and structural analysis revealed that farm features attracting insects were associated with higher bat activity, while the surrounding habitat showed little effect. Crucially, we found frequent absence of physical barriers preventing contact between bats or their droppings and pig enclosures, increasing exposure risks. Focusing on the most common bat species, we detected active CoV circulation in *P. kuhlii*, including colonies located near pig facilities. Two distinct CoV species were identified in *P.*

**Data availability statement:** Consensus sequence have been submitted to NCBI GenBank under accession numbers PV420065-PV420123. Row data from the metagenomics sequencing have been submitted to SRA under accession number PRJNA1234035.

**Funding:** The present work was supported by the First International ICRAD call under grant agreement N◦ 862605, ID 95 ConVErgence.

**Competing interests:** The authors declare no competing interests.

*kuhlii*, suggesting potential for viral recombination. CoVs were detected throughout the active season, with amplification peaks in May and August. Phylogenetic analysis indicated that pigs could be exposed to at least eight bat CoV species in Italy. Notably, CoVs appeared to be shared between *P. kuhlii* and *H. savii*, further increasing recombination risks. Our study outlines a potential transmission route of bat CoVs to swine and highlights key risk factors, including farm structures, biosecurity gaps, bat species involved, viral diversity and seasonal patterns of virus circulation.

## Introduction

The rising incidence of human emerging infectious diseases (EIDs) originating from wild animals can be traced back to a common set of anthropogenic drivers, such as climate change, urbanization, and land-use conversion, which alter the distances and contact rates between wildlife and humans [1–3]. Indeed, the Anthropocene is facing increased encroachment because wild animals are forced into human environments due to reduction, fragmentation or disturbance of their natural habitats and, conversely, people are progressively breaching wild environments for work or entertainment (i.e., hiking, scouting and hunting) [4–6]. In addition, the establishment of pastures, plantations or livestock farms near forest edges can facilitate the spillover of wild pathogens in domestic animals that, by having a closer proximity to humans, might easily act as bridge hosts for future human infection, especially when reared in intensive settings [7–12]. This is clearly exemplified by the role of domestic animals in several human epidemics caused, for example, by orthomyxoviruses [13], coronaviruses (CoVs) [2,14–17], and henipaviruses [18]. In this context, the emergence of diseases is likely just the tip of the iceberg of increased spillover events that remains unnoticed in both domestic animals and humans. Indeed, serological evidence showed how people living in the agricultural areas of Southern China were already exposed to SARS-like coronaviruses before one of these viruses was able to jump into the human population, causing the pandemic of Coronavirus Disease 2019 (COVID-19) [19]. While urbanization and other anthropogenic pressures are known as a threat for biodiversity, animals can be attracted to and thrive in anthropogenic environments when, based on their biology and phenology, are able to exploit the consistent resources that in these settings tend to be stable over time and space [20]. Species included in this environmental micro niche are more likely to act as source of pathogens, due to higher interaction with humans and domestic animals, therefore, understanding the actors and the triggers of these interactions might then be a crucial step in the risk assessment for the emergence of novel diseases.

Among wildlife, bats are known to be host of a wide viral diversity, including species that are well known for their zoonotic and epidemic potential, such as coronaviruses [21–23]. The potential appeal of anthropogenic resources for bats is strictly dependent on the availability of feeding grounds, drinking sites and refuges in the natural environment [24,25]. As natural roosting sites have become increasingly scarce due to habitat loss and changes in land use, buildings have gained significant

importance for bats [20]. Human-made structures (e.g., crevices in walls, holes under tiles, bridges and gutters) can mimic the features of natural roosts like cliffs, caves, or trees. Furthermore, human-created water sources like artificial ponds, reservoirs, cattle troughs, or swimming pools, located within or near urban environments, drinking opportunities and food resources [26–28]. In fact, bats are particularly vulnerable to dehydration due to their high body surface area in relation to body volume, and the presence of bare wing membranes, which facilitate transpiration [29,30]. This phenomenon is particularly critical in arid or semiarid regions, as well as in the Mediterranean, where water availability is often limited either permanently or seasonally. As for animals in general, the adaptability to human environment and the actual use of artificial structures is different between bat species that, in turn, might have different likelihood of being the host of the next epidemic human pathogen. In this context, focusing on viruses associated with synathropic bat species might be a way to prioritize research in the field of disease ecology.

In this study, we wanted to move forward the prioritization of bat pathogens that should be monitored for their emergence in Italy by identifying bat species that most interface with humans, and describing the viral species circulating in these hosts. We focused on coronaviruses, because of their frequency and diversity in bats and because history already showed us their zoonotic and epidemic potential [23]. Coronaviruses are the largest family of positive-sense, single-stranded RNA viruses, classified within the *Nidovirales* order and officially divided into four genera: *Alphacoronavirus* (α-CoV), *Betacoronavirus* (β-CoV), *Gammacoronavirus* (γ-CoV), and *Deltacoronavirus* (δ-CoV) [23]. In addition, divergent viruses have recently been described in mustelids and may represent, upon approval from ICTV, two novel genera: *Epsiloncoronavirus* [31] and *Zetacoronavirus* [32]. A hallmark of CoVs evolution is their frequent host shifting, occurring between animals and, occasionally, from animals to humans. This dynamic has led to serious epidemics, with notable recent examples including COVID-19, Middle East respiratory syndrome (MERS), and severe acute respiratory syndrome (SARS), associated respectively with the spillover of SARS-CoV-2 in 2019 [33], MERS-CoV in 2012 and SARS-CoV in 2002 [34–36]. Following the evidence that most human coronaviruses emerged after a passage in intermediate hosts, we used bioacoustics to investigate which bats interact more frequently with animals in intensive productions, focusing on swine that, being the domestic species associated with the highest number of species-specific coronaviruses, could more easily act as bridge for human infection. Based on the bat richness and frequency in piggeries we performed a landscape analysis to define the structural, environmental, and epidemiological factors driving the richness and activity of bats at the interface with swine that could facilitate virus interspecies transmission. Finally, we aimed at describing the ecology of coronaviruses circulating at this interface by performing active surveillance to provide preliminary data on prevalence and seasonality and whole genome sequencing to enhance our knowledge on strains circulating locally. Building from the extensive evidence that CoVs tend to be species-specific [37] we also used bat richness to infer the viral species that could be present at this interface and might have been overlooked during surveillance.

While the silent circulation and amplification of viruses spilled over domestic animals could facilitate secondary transmission to humans [38], coronaviruses could also be a serious threat for animal health, as shown by the economic and sanitary consequences associated with the emergence of porcine epidemic diarrhea virus (PEDV) and swine acute diarrhea syndrome-coronavirus (SADS-CoV) [37], both showing phylogenetic relatedness with viruses circulating in bats [39,40]. In this context, understanding the ecology of coronaviruses circulating at the interface between bats and pigs is crucial for public health as well as for animal health.

## Materials and methods

### Interface between bats and pigs in North Eastern Italy

The study wanted to investigate bats populations circulating in swine farms and identify ecological, structural and landscape characteristics that drive the biodiversity of bat species and the level of activity in swine farms, using North Eastern Italy as a study site that could be representative of intensive swine farming.

**Study area and bioacustic analyses.** We performed a bioacoustics analysis to investigate the presence of bats in 14 pig farms from Veneto and Friuli Venezia Giulia regions in North Eastern Italy, selected to represent the different size, production and management strategies implemented in the area (Fig 1). This region is characterized by medium to high density of pig farms, primarily involved in intensive production systems aimed at rearing pigs for the heavy production of traditional dry-cured ham, particularly the Designation of Protected Origin (DPO) San Daniele ham. Farms have been randomly labelled 1–14 so that no identification of exact sites is possible. The study was performed one on each farm between April and October in 2021 and 2022. For each farm, the moment of sampling was organized according to logistics and the availability of farmers, and was limited by biosafety measures implemented during the winter epidemics of Highly Pathogenic Avian Influenza (HPAI) H5N1 [41] to reduce possible cross-species transmission events. Later sampling was hampered by the introduction of African Swine Fever in Italy in early 2022 [42].

Bioacoustics is a non-invasive tool to monitor bats, consisting in the recording and analysis of the echolocation calls that most bat species produce for orientation and prey detection. In each study site, we positioned one automatic bat recorder (Song Meter SM4 Acoustic Recorder, Wildlife Acoustics) inside the farm perimeters, choosing areas that could promote the presence of bats thanks to linear structures and landmarks in the territory, such as flight corridors, water bodies or sewage tanks, though distant from light sources. In case bat colonies were found in buildings inside farms, the device was positioned to avoid over-sampling of resident individuals. We programmed the devices to record all bat sounds between sunset and midnight when all species leave their shelters to commute and feed and left them on site for a minimum of one week. Depending on the level of bat activity and background noises coming from farming practices, the battery lasted between three and seven days, providing data for an average of five. Raining days were discarded from analyses as bats are known to limit their activities whenever weather conditions are not optimal [43,44]. We analysed bat sounds using BatSound 4.12 (Pettersson Elektronik AB, Uppsala) and Raven Pro 1.5, with no automated algorithms, using reference calls following Russo & Jones [45]. We identified bats to species level including those which have a high overlap among signals, with exception made for the genera *Myotis* which were pooled and classified as *Myotis* sp.,

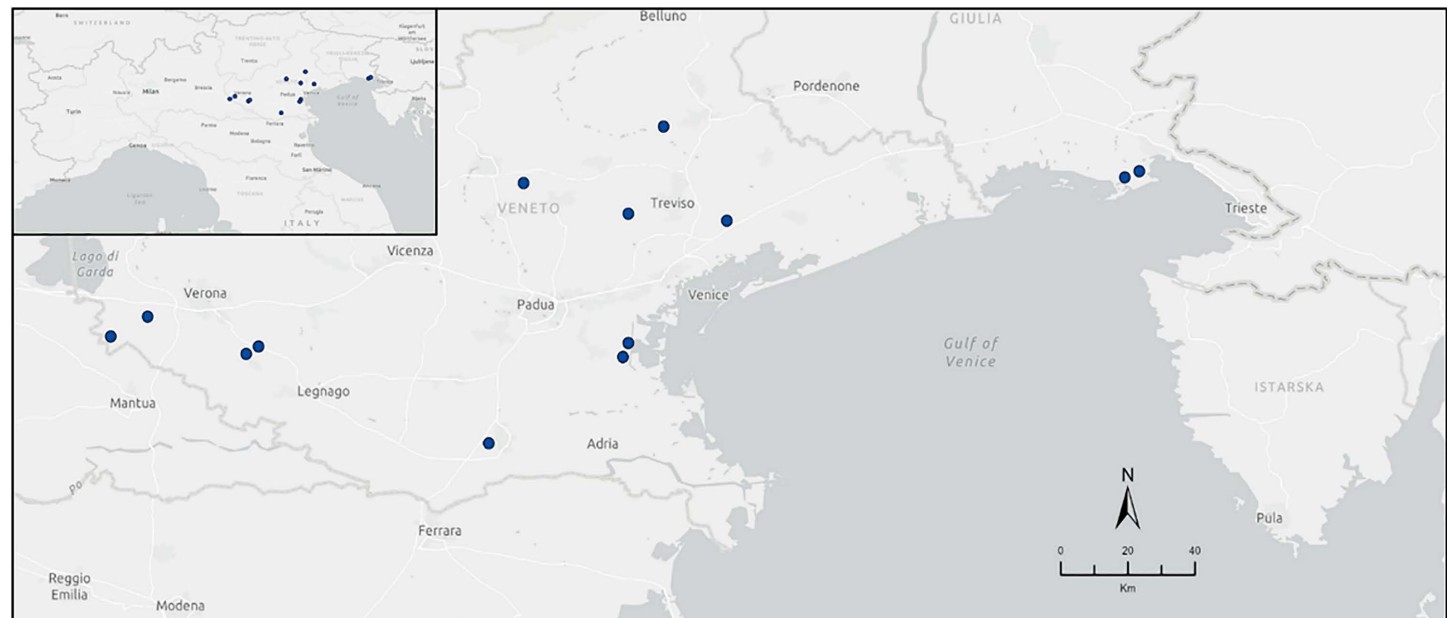

**Fig 1. The fourteen sampling sites selected to perform the study.** The capture in the bottom left of the figure is representative of the Italian regions considered in the study. Dots in blue represent the location of the pig farms. The map has been produced using ArcGIS Pro (ESRI ©).

double-checking signals that were disturbed or misinterpreted. We distinguished calls based on their scope in echolocations calls (specific calls for the echolocation), feeding buzzes (species-specific calls performed during hunting phases), and social calls (species specific calls performed during commuting phases with conspecifics). We used these data to establish the activity and the richness of bats in each sampling site, the former calculated as number of bat passes per night, determined as average of all available recording days, and the latter as total number of bat species recorded on each farm. Finally, we explored the feeding and the social activity of bats in general and for each species by calculating the buzz-ratio, defined as the percentage of feeding buzzes over the total passes occurring in each farm and the percentage of social calls over the total passes occurring in each farm.

**Landscape analyses.** We described the structural and ecological features of our study area using different approaches. We described the configuration of each farm by investigating pre-defined features known to facilitate or discourage the presence of bats based on the available literature. We collected information on these features on a standardized data sheet that we filled during our surveys either by direct evaluation or gaining specific information from farm owners or veterinarians (see S1 Table of the supporting informations). We considered i) general features, including the farm area and the age of the construction of the buildings; ii) features known to attract insects, including the number of pigs, the presence of sewage tanks, unused rooms, the number of light sources in the farm or in close proximity and the presence of irrigation canals; iii) the presence of structures known to discourage the presence of bats (in particular the level of noise); iv) features that provide shelter to bats, such as shutters, space behind gutters, crevices, broken bricks and holed trees and v) features that influence the likelihood for contacts between bats or bat faeces and swine. This included open windows or doors, which could allow entrance of bats, and the lack of grills/nets at the windows and air intake inlets, which could allow contamination with bat faeces from the outside. We did not asses the actual entrance of bats or presence of bat guano in study sites due to technical and logistical challenges to obtain standardized data. However, if we noted the actual presence of guano, indicative of individuals roosting in the farm, we recorded the information and collected faeces to identify the bat species with a genetic approach and to screen for the presence of CoVs, as described below.

We described the habitat type within a buffer of 300m radius around the position of each bat detector using a geodatabase in ArcGIS Pro (ESRI ©), considering this range as a proxy of foraging areas for bats [46,47]. Inside each buffer we digitized all landscape categories and divided into patches considering anthropogenic structures, agriculture, woods, and water bodies using Google Earth engine as data source. For each buffer we calculated the percentage occupied by each landscape category and we calculated the number of patches from each landscape category in each buffer (used as a proxy of environmental heterogeneity) [48]. We also measured the Euclidean distance from the nearest forest edge, the Euclidean distance from any water source (see S1 Table of the supporting informations).

**Bats' ecological niche in swine farms.** We paired results from bioacoustics and landscape analyses to investigate parameters that can influence the biodiversity and frequency of bats in pig farms from North Eastern Italy. We aimed to quantify the relationship between the habitat features surrounding the pig farms and i) bat species richness, ii) overall bat activity and iii) activity of *P. kuhlii,* being the most frequently recorded species in the area. Statistical analyses were performed in the Rstudio (v4.2.2; https://www.r-project.org/).

Predictors were selected among all the structural and ecological parameters collected (described in S1 Table of the supporting informations) based on a univariable regression analysis that was run preliminary for each of the three analyses (response variables: overall activity, *P. kuhlii*-specific activity, bat richness), then we ran multivariable regressions relying on multi-model approaches [49].

To facilitate interpretation of parameters estimates, non-binary variables (distances, farm size, number of patches and pigs) were normalised to range between 0 and 1 (1 being the maximum observed value). All models assumed that the response variables followed a negative binomial (NB) distribution, and the NB regressions relied on the mgcv R-package [50]. Each tested predictor was included/excluded from the multivariable analysis based on Akaike Information Criterion

(AIC). To increase the sensitivity of our analysis, we applied a weak criterion, i.e., comparing models with/without the single predictor, a single unit decrease in AIC was sufficient to warrant predictor inclusion. This allowed us to define the set of 'selected predictors' to be used in the multivariable analyses [49] (see S2 Table in supporting information).

We ran multivariable regression evaluating the complete set of models including up to three selected predictors but, to assess the robustness of the results, the multivariable analyses were repeated restricting models also to a maximum of two and four predictors. The restriction to three variables stemmed from avoiding over-parametrisation of models, given the sample size. We accounted for the fact that the parameters related with the habitat types were always complementary within buffer zones and added to 1 (i.e., 100% coverage of the buffer). This concept was always true for the parameters *anthropic structures, agriculture* and *wood*, while the proportion of *water bodies* was minimal, accounting for 1.4% of the land-use on average. For this reason, we analysed the variable *water bodies* independently while considering the other habitat types as a three-dimensional single variable, called land-use. Whenever this variable was included as predictor in the multivariate analysis, we included a combination of all three landscape categories expressed as a ternary plot using the Ternary R-package [51] (see S1 Fig in supporting informations). This facilitated the interpretation of the results, as predictions based on increase in one variable should be associated with a decrease in the others. Given the high correlations between the land-use and distance to water (and wood), we excluded models that included both land-use and distance to water (and land-use and distance to wood). The multi-model analyses was custom built with minimal reliance on the AICcmodavg R-package [52] (i.e., only used to compute model-averaged coefficient estimates and 95%CI as described below). For each final model, we extracted estimated coefficients, AICc (i.e., correcting for small sample size) and Nagelkerke pseudo-Rsq [53]. AIC weight was calculated for all models evaluated and used to infer each variable Relative importance (Ri: sum of the Akaike weights of models including the variable of interest). Model average of each variable was extracted based on all models, and 95% confidence interval (CI) based on unconditional standard errors for the model-averaged estimate. Model average estimate and 95%CI relied on the AICcmodavg R-package [52].

## Bat coronaviruses at the bat/swine interface

The study wanted to enhance our knowledge on the coronaviruses circulating at the interface between bats and swine reared in intensive settings, including the prevalence, seasonality, genetics and diversity. The work is part of a bigger study where swine has been screened for coronaviruses as well (Festa et al., in preparation). As no bat viruses where found these data were not included in this paper.

**Prevalence and seasonality of CoVs found in *P. kuhlii* in Italian piggeries.** This objective was focused on coronaviruses associated with *P. kuhlii* that was the bat species founded more frequently showing highest activity rates from the bioacoustics investigation described above, as previously reported [54]. In addition, three colonies were spotted within piggeries 4, 6 and 10 during inspections, allowing for the collection of guano. This environmental pooled faecal material was used for genetic identification of the roosting bats through the amplification and sequencing of the Cytochrome Oxidase I (COI) [55], using the BOLD-System software available online (http://www.boldsystems.org). In addition, all pools were screened for coronaviruses. Among these three sites, we selected colony A within farm n. 10 for longitudinal surveillance, because easy access granted by the farm owner. In addition, the fact that this colony was already spotted and found out positive for CoVs in previous investigations [54] supported its stability over time that would have been necessary for longitudinal screening. In order to increase data for this investigation, we performed active surveillance of two further colonies of *P. kuhlii,* colonies B and C, located in a house crevice and a bat box in a peri-urban area of the Veneto Region. These colonies were selected in private properties upon confirmation of the host species and virological positivity to coronaviruses. We screened all colonies six times between 2021 and 2022, collecting faecal pools from below the colony. In colonies B and C, we were able to clean the surface of the soil beneath the roosting site a couple of days before each sampling session, in order to remove environmental dirt and old faeces and collect 20 fresh pellets from each sampling time, that could be considered as representative of single individuals. On the other hand, the

collection of fresh environmental samples was complicated in colony A by environmental (very hot or rainy days prior to our fieldwork campaigns) and management (cleaning of the area) conditions within the swine farm that were out of our control. In this location, we collected only visibly fresh available faecal pellets and attempted bat capture to obtain better samples in terms of quality and quantity. In addition, bats were captured using misnets positioned in front of the gutter. Live sampling of bats have been performed under manual restraint following the best practices of the field to minimize disturbance and avoid any suffering, upon ethical approval of the Italian Ministry of Environment and Energy Security and the Ministry of health, upon disclosure of Directive 91/43/CEE Habitat (based on ISPRA assessment 9008 of 25th February 2021). In colony A, we were unable to standardize the number of samples to 20 as for the other settings.

In all cases, we stored faecal samples in tubes with 1 ml of RNAlaterTM stabilizing solution (Invitrogen, Massachusetts, USA) aimed at preserving the genetic material at room temperature before submission to the laboratory. RNAlaterTM was removed before freezing samples at −80°C.

We extracted nucleic acids from faecal pools using the QIAamp®Viral RNA Mini kit (QIAGEN, Hilden, Germany). We investigated the presence of CoVs using a nested RT-PCR showing a broad spectrum across all CoV's genera [56]. All amplicons were sequenced using a Sanger approach and analysed using a maximum likelihood (ML) phylogenetic approach to investigate correlations with strains already described in the literature. This analysis is detailed in the following paragraph titled "*Estimate of the bat coronavirus' diversity in swine farms from North Eastern Italy*". We performed a Kruskal-Wallis test to examine the potential difference in terms of percentages of positivity noticed between colonies [57]. For population B and C, whose sampling strategy was effectively standardized over time, we performed a K proportion test to compare the results obtained in the different months of surveillance. We performed statistical analyses using R.

**Genetics of Italian bat CoVs that could circulate at the interface between bats and pigs.** While several studies describe the partial *RdRp* of bat CoVs, few works succeeded in obtaining the whole genome sequence of strains circulating in Italy in the bat species of interest, namely the one found to circulate most in piggeries during our bioacoustics survey.

In order to better characterize the taxonomy of viruses from our territory, we attempted whole genome sequencing of CoV positive samples detected in *P. kuhlii* and *H. savii,* the two bat species that showed higher frequency and activity in the bioacoustics analyses performed in our study. Positive samples to be sequenced were either obtained from our active surveillance or in the framework of other studies. In particular, we exploited samples from the extensive passive surveillance performed in Italy in response to COVID-19 epidemics [58] that is still performed as routine in selected wild mammals at the Istituto Zooprofilattico Sperimentale delle Venezie, Italy. In the specific case of bats, this screening is performed on lungs and intestines derived from carcasses submitted at the National Reference Centre for Rabies as mandatory screening against *Lyssavirus*, whose species is always confirmed through the amplification and sequencing of the Cytochrome Oxidase I (COI) [55]. We retrieved all CoV positive samples found in *P. kuhlii* and *H. savii* and selected the ones to be submitted to WGS based on data from RNA integrity assessed using the 2100 Bioanalyzer system and the RNA 6000 Nano or Pico Total RNA Assay (Agilent Technologies, Santa Clara, CA, USA). Host ribosomal RNA depletion was applied starting from 20–30 ng of total RNA input using the NEBNext® rRNA Depletion Kit (Human-Mouse-Rat and bacteria) according to the manufacturer's instructions (New England Biolabs Ipswich, MA, USA). Double strand cDNA was synthesised applying the Sequence-independent, Single-Primer Amplification (SISPA) protocol [59]. Illumina DNA PREP kit (Illumina, San Diego, CA, USA) was used to produce sequencing libraries according to the manufacturer's recommendations. Sequencing was performed on the Illumina Miseq instrument in 2 × 250 bp Paired-End mode (Illumina, San Diego, CA, USA).

Raw data were filtered by removing: i) reads with more than 10% of undetermined ("N") bases; ii) reads with more than 100 bases with a Q score below 7; iii) duplicated paired-end reads. Remaining reads were clipped from Illumina adaptors with scythe v0.991 (https://github.com/vsbuffalo/scythe) and trimmed with sickle v1.33 (https://github.com/najoshi/sickle). Reads shorter than 80 bases or unpaired after previous filtering were discarded. Taxonomic assignment of high-quality reads was carried out using the Basic Local Alignment Search Tool (BLAST 2.10.0+) alignment against the integrated NT

database (version 23 February 2020) and Diamond v0.9.17 [60] alignment against the integrated NR database (version 23 February 2020). Alignment hits with e-values greater than $1\times10^{-3}$ were discarded. The taxonomical level of each read was determined by the lowest common ancestor (LCA)-based algorithm that was implemented in MEGAN v6.18.50 [61]. For the reconstruction of the complete genome, a subset (0.5%) of reads taxonomically classified as belonging to *Coronaviridae* family was selected and de novo assembled using IDBA-UD v1.1.1 [62] with the multi-kmer approach using a minimum value of 24, a maximum value of 124 and an inner increment of 5. To ensure that all the viral reads would be represented in the main assemble, the longest contig produced was subsequently used as reference for a reference-based assembly. High quality reads were aligned against such reference genome using BWA v0.7.12 and standard parameters [63]. Alignments were processed with SAMtools v1.6 to convert them in BAM format and sort them by position [64]. SNPs were called using LoFreq v2.1.2 [65]. According to LoFreq usage recommendations, the alignment was first processed with Picard-tools v2.1.0 (http://broadinstitute.github.io/picard/) and GATK v3.5 in order to correct potential errors, realign reads around indels and recalibrate base quality [66]. LoFreq was then run on a fixed alignment with option "--call-indels" to produce a vcf file containing both SNPs and indels. From the final set of variants, indels with a frequency lower than 50% and SNPs with a frequency lower than 25% were discarded. To produce the consensus sequence, we changed the reference genome in agreement with the following rules: i) for a position j, if coverage was not enough to reliably call variants, we added an "N" base; ii) for a position j, if coverage was enough to reliably call variants but no SNP were called, we added a reference genome base at position j; iii) for a position j, if coverage was enough to reliably call variants and at least one SNP were called, we added the nucleotide using the IUPAC nucleotide code (http://www.bioinformatics.org/sms/iupac.html) according to the bases present. Finally, high quality reads were re-aligned with BWA against the consensus sequence produced; we performed a visual inspection of the alignment with tablet v1.14.10.21 and, if required, manually revised the consensus sequence based on this alignment [67].

We used reference whole genomes with highest identity as identified by BLAST analysis to annotate and characterize sequences obtained within this project. We identified relevant CoV genes within our genomes using the 'Annotate & Predict' function of Geneious Prime. We then confirmed the subfamily and species of our strains by performing phylogenetic analyses and by calculating pairwise evolutionary distances for conserved domains of the RdRp, namely ADRP, nsp5 (3CLpro), nsp12 (RdRp), nsp13 (Hel), nsp14 (ExoN), nsp15 (NendoU) and nsp16 (O-MT). As suggested by the International Committee on Taxonomy of Viruses (ICTV) and performed in recent taxonomic studies, we used 90% aminoacidic identity and phylogenetic clustering to associate our viruses with their CoV species [68]. In order to investigate major recombination events, we compared phylogeny based on target domains of RdRp and structural M, E, N and S proteins.

**Estimate of the bat coronavirus' diversity in swine farms from North Eastern Italy.** To investigate the diversity of coronaviruses that could circulate at the interface between bats and pigs in the study area, we performed a phylogenetic analysis based on published sequences to identify all viral strains that have been associated with the bat species detected in our acoustic sample. We downloaded from GenBank sequences of partial RNA dependent RNA polymerase (RdRp) representative of all strains associated with bat species found during our bioacoustics survey, sampled from different locations and years. We prioritized strains for which accurate taxonomy could be defined by available whole genomes. The *RdRp* from the new sequences described in this study were also included in the dataset, together with the *RdRp* from strains of bat CoVs showing 80–100% identity with them, identified by BLAST analysis. We aligned sequences of bat coronaviruses with representative strains of swine coronaviruses in order to investigate the phylogenetic relatedness between CoVs found in the two host types. We aligned the partial RdRp (381base pairs) of 500 sequences of coronaviruses founded either in bats or pigs using the online tool Mafft, selecting the settings "G-INS1" and leaving the other parameters as default (https://mafft.cbrc.jp/alignment/server), and built a Maximum Likelihood (ML) phylogenetic tree using PhyML (version 3.0) implemented in the Seaview program (Lyon, France), setting a GTR + G4 substitution model and an algorithm for supporting SH-like clusters. The tree was edited using the iTol program (https://itol.embl.de/) to highlight the taxonomy of viruses identified in different bat species and swine.

We assessed the diversity of CoV species circulating at the interface between bats and Italian pigs, by calculating the genetic aminoacidic distances within and between clusters identified through phylogeny using MEGA6, considering 90% aminoacid identity as threshold to classify sequences as probable members of the same species. Whenever needed, we used available complete genomes for each cluster to confirm taxonomy. We used this analysis to calculate the number of CoV species currently described in bats that could come into contact with pigs in our setting.

## Results

### Interface between bats and pigs in North Eastern Italy

**Bioacoustics analysis.** We performed our field investigation in 14 pig farms sheltering between 100 and 8000 pigs, two of those are located in the Friuli Venezia-Giulia and twelve in the Veneto region (Fig 1). Overall, we obtained 86 days of acoustic sampling, recording 4587 bat calls. Most recorded calls were echolocation signals, while feeding passes and social calls were limited to a maximum of 5.14% and 9.31% of the samples respectively (see S3 and S4 Tables in supporting informations).

We estimated bat activity from the 4587 echolocation calls, as the main number of passes per night. We recorded an average of 45 passes/night across different farms (s.d. = 64,41; overall range: 13–211). In all farms, the highly synanthropic *P. kuhlii* was the most active species, with passes ranging from 24 to 1062 (mean 200 passes), followed by *P. pipistrellus,* with 2–381 passes (mean 48,21) and *H. savii,* with 2–41 (mean 6,71) passes.

In total, we recorded at least 8 different species (i.e., *Pipistrellus kuhlii, Pipistrellus pipistrellus, Hypsugo savii, Eptesicus serotinus, Myotis* sp*., Nyctalus leislerii, Rhinolophus hipposideros* and *Rhinolophus ferrumequinum*). Due to the considerable overlap in the acoustic signals of *Myotis* species, we were unable to identify them at the species level, possibly, with more than one member of the genus *Myotis* possibly circulating, thus increasing the total amount of species. The species richness differed between farms from one to six (average richness of 2 species detected per single farm) (s.d. = 1,25) (Fig 2). The most abundant bat species were *P. kuhlii* and *P. pipistrellus*, present in all and 13 out of 14 sampled farms, respectively. Those species were followed by *H. savii* (present in 9 farms) and *E. serotinus* (in 5 farms);

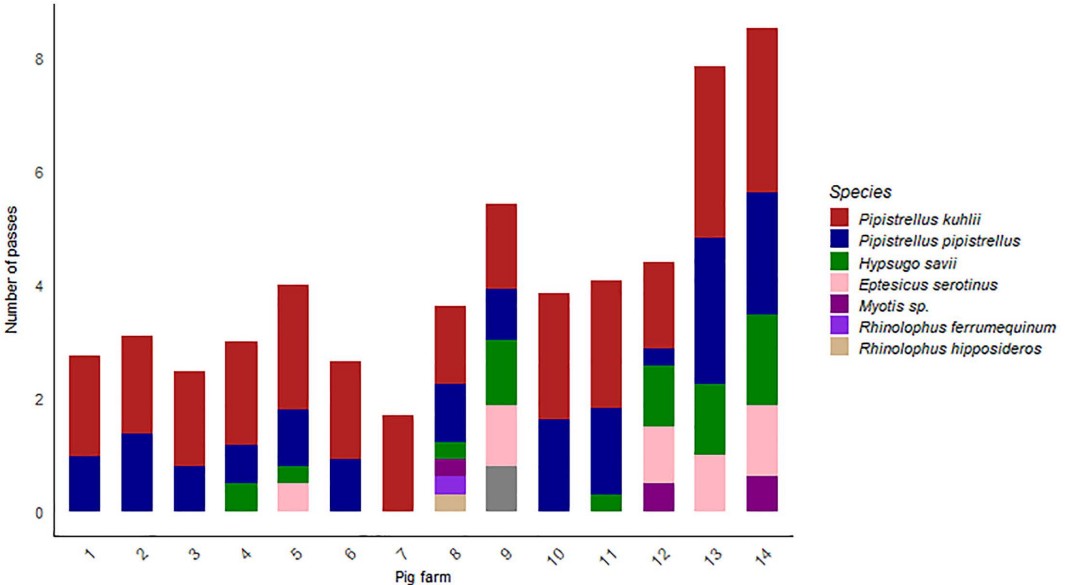

**Fig 2. Bat activity and relative richness in the 14 pig farms under study.** Activity per species was calculated as the average number of species-specific passages across the effective sampling days per farm.

the species *N. leislerii, R. hipposideros*, and *R. ferrumequinum* were detected only occasionally. Members of the genus *Myotis* were recorded in three different farms; we were able to discern the occurrence of one *Myotis* species respectively in farms and 8 and 14 but relatively to farm 12 we were not able to assess the presence of a single species.

**Landscape analysis.** The landscape analysis allowed us to describe the structure, the ecological parameters of farms and the type of habitat surrounding them (see Fig 3 in the main text and S1 and S5 Tables in supporting informations). Farms included in our sample were either medium or large, with mean area of around 28.000 m², ranging from 9.000 to 65.000 m², hosting between 100 and 8000 pigs (mean 3.150 pigs). The majority of farms included one or more sewage tanks (9/14) that are known to attract insects and, in turn, to potentially attract bats, especially when not covered. In addition, four farms showed elevated number of insects within empty rooms, mostly found in older abandoned buildings within renovated farms. In some cases farms were very much illuminated with up to 5 sources of lights in the whole area, potentially shifting both the insect and the bats community toward those lamps [69]. Old and new farms were almost equally represented in the sample. Regardless, most of the buildings lacked features known to provide good shelters for bats such as space behind gutters, crevices/broken bricks or shutters, which were present in 6, 5 and 4 out of 14 farms respectively. Holed trees were only seen in three farms. Bat roosts as testified by the presence of large amount of guano was noticed in three farms as well (n 4, 10 and 11), with bats always positioned behind gutters. In one case, the colony was located outside the pigs' shed, in the other two the gutters were positioned in administrative buildings. All bats were confirmed as *P. kuhlii* through genetic analyses of the guano. Only the guano collected outside the animal shed of farm 10 turned out positive for CoVs. This population was included in this study for the active surveillance of coronaviruses as "Colony A".

The landscape analysis allowed us to describe for all farms a similar habitat, dominated by agriculture environments (with a mean value of 218.499 m²), followed by anthropogenic structures (45.198 m²) and wood (23.783 m²), and with a very small percentage of remaining patches of water (4.521 m²). Table 1 shows how farm 6 could be considered an outlier from our sample, being the only extensive farm where pigs of all ages were reared outdoors in a woody environment. Other values calculated in this analysis were the distance from patches of wood and water that mostly depends on the location of the farms; as a mean value piggeries are usually distanced from the wood 292 m (s.d. = 273 m) and from the water 511 m (s.d. = 395 m). Generally, areas around farms showed a nice degree of habitat heterogeneity, with more than 10 different patches detected in 6 out of 14. Low heterogeneity of less than five patches was detected only in three farms.

**Bats' ecological niche in swine farms.** We run a multivariable analyses selecting predictors based on univariable regression. Results from these analyses were highly robust using up to two, three or four predictors at time, with consistent results among the three analyses (see Table 1 in the main text and S6, S7 Tables and S2 Fig in supporting informations).

Considering variables related to the configuration of the farm, the presence of empty rooms was positively and significantly associated with all the dependent variables of bats' richness and activity (Table 1). Its relative importance was also consistently estimated as the highest, reaching between 78% and 98%. The number of pigs in the farm was also associated with increased overall activity and *P. kuhlii* activity, with relative importance reaching 71% and 90% respectively. To a lesser extent, newly built buildings and the presence of illumination were also associated with an increased activity, albeit with lower relative importance, and the association was neither found for *P. kuhlii* activity. The distance to water was associated with richness only, with intermediate relative importance (45%). The proportion of landscape categories used for agriculture, urban and wood were consistently associated with decreases across the three dependent variables, albeit weakly (lower relative importance and 95% CI overlapping 0). Distance to wood showed a similar weak relationship with overall activity only.

## Viruses at the bat pig interface

**Prevalence, seasonality and genetics of Italian bat CoVs at the interface with pigs.** During our surveys we detected piles of guano indicative of bat roosts within farm n. 4, 10 and 11, from which we collected pools for genetic

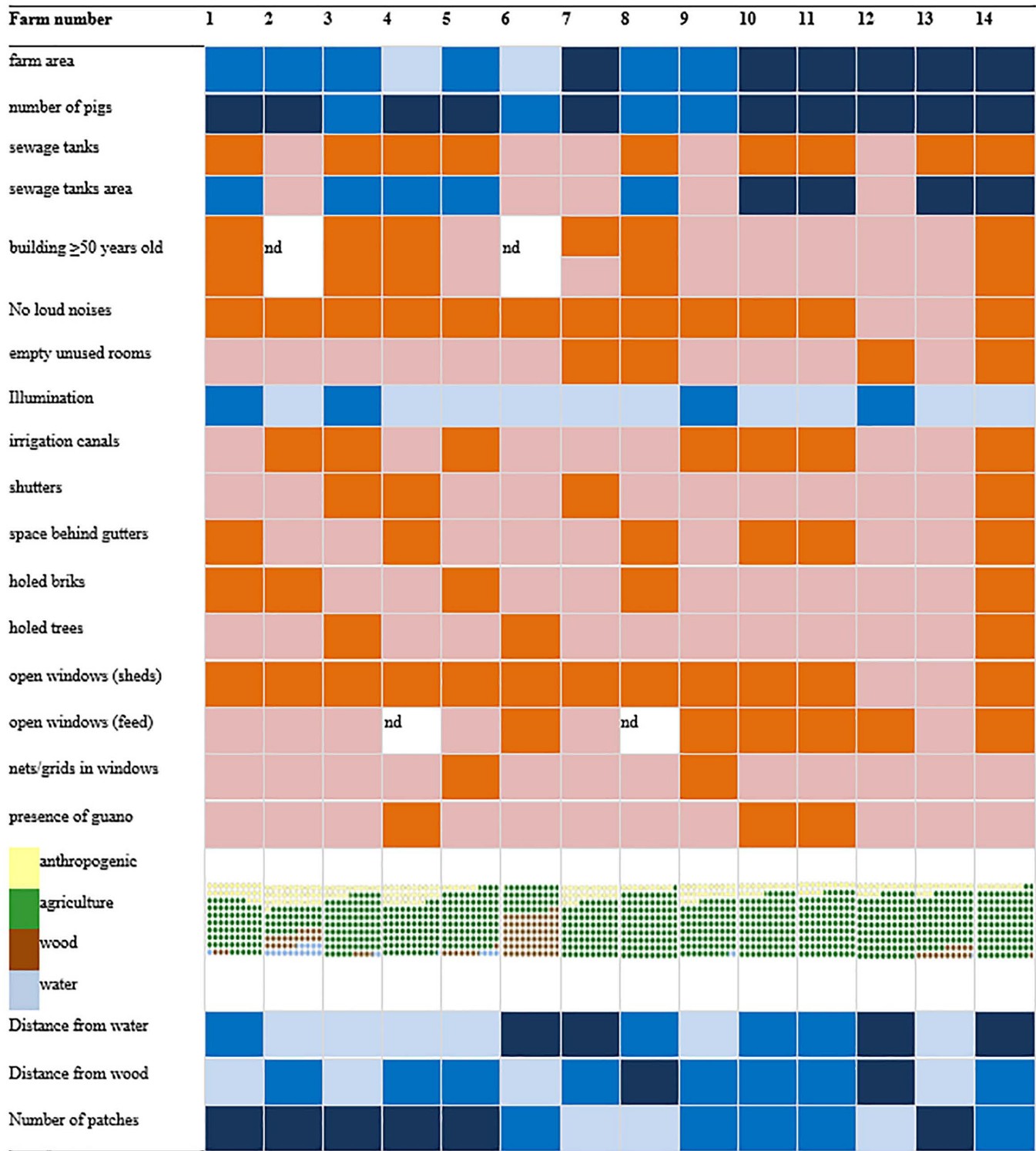

**Fig 3. Descriptive results of the landscape analysis.** Cells are coloured in orange for yes and pink for no. For quantitative data, cells are coloured in three shades of blue (light blue -lb-; blue -b-; dark blue -db-) upon increasing values. Farm area (m²): lb < 10.000, medium 10.000-30.000, dark >30.000; Number of pigs: lb < 30, b 30-1000; db > 1000; sewage tanks (m²): lb < 100, b 100-1000, db > 1000; n of light sources: lb ≤ 1, b 2-5, db > 5; distance from water/wood (m): lb < 150, b 150-500, db > 500; n of patches: lb < 5; b 5-10, db > 10.

**Table 1. Summary of model-averaging coefficient when predicting either 'overall activity'. 'P. kuhlii activity' (P.k.), or bat species 'richness'.**

| | Variable | Overall activity Coefficient (95%CI); Ri | P.k. only Activity Coefficient (95%CI); Ri | Richness Coefficient (95%CI); Ri |
|---|---|---|---|---|
| Farm related | Building age | 0. 93 (0.27, 1.58); 0.64 | | |
| | Farm area | | | |
| | Number of pigs | 3.1 (0.46, 5.75); 0.71 | 2.8 (1.49, 4.12); 0.9 | |
| | Presence of empty rooms | 1.55 (0.49, 2.6); 0.78 | 1.02 (0.46, 1.58); 0.98 | 0.41 (0.13, 0.7); 0.92 |
| | Presence of illumination | 1.25 (0.34, 2.16); 0.35 | | 0.26 (−0.02, 0.55); 0.35 |
| | Presence of irrigation canal | | | |
| | Presence of space behind gutters | | | |
| | Presence of shutters | | | |
| | Presence of holed trees | | | |
| | Size of sewage tank | | | |
| Landscape related | Agriculture (proportion) | −6.25 (−15.56, 3.06); 0.16 | −2.06 (−12.06, 7.95); 0.22 | −0.53 (−6.57, 5.51); 0.16 |
| | Urban (proportion) | −11.83 (−24.65, 0.99); 0.16 | −6.49 (−18.94, 5.96); 0.22 | −3.09 (−10.8, 4.63); 0.16 |
| | Wood (proportion) | −6.39 (−16.61, 3.84); 0.42 | −2.9 (−13.34, 7.54); 0.22 | −1.26 (−7.74, 5.23); 0.16 |
| | Water (proportion) | | | |
| | Distance to water | | | −0.7 (−1.39, −0.01); 0.45 |
| | Distance to wood | −1 (−2.47, 0.47); 0.26 | | |
| | Number of patches | | 1.12 (0, 2.24); 0.08 | |

Coefficients, 95% Confidence Intervals (CI) and Relative importance (Ri) associated with each variable considered. When no estimate is provided, the variable was excluded from the analysis based on the preliminary univariate regression. Blue (red) shading indicates positive (negative) relationships, while darker shading indicates estimate for which the lower (upper) 95% CI bound did not overlap with 0 (i.e., this can be interpreted as a statistically significant impact under an hypothesis-testing framework). The table presents results when including up to 3 predictors (see S6 and S7 Tables and S2 Fig in supporting informations for estimates when allowing up 2 or up to 4 predictors).

and virological analyses. Genetic analyses confirmed that all colonies belonged to the species *P. kuhlii.* Among these, only the population roosting in farm n.10 yielded positive results for CoVs and was selected as colony A for subsequent longitudinal screening. On the other hand, the two further colonies investigated longitudinally in this study for the presence of CoVs (colonies B and C) were selected on private properties unrelated with pig farms upon confirmation of *P. kuhlii* as host species and positive screening for CoVs (see S8 Table in supporting informations). From colony A, we obtained 113 samples, including a mix between individual samples obtained through the capture of animals (n = 18) and pooled samples collected from beneath the colony (n = 95, ranging from 2 to 27 sampling occasion, with a mean collection of 16 samples per sampling occasion). On the other hand, we were able to carry out the systematic collection of 20 samples for each sampling occasion of colonies B and C, obtaining 120 faecal pools for each location. Out of 353 samples, 55 were positive for coronaviruses (15.6%). Due to limited sample size and lack of data about the population dimension we could not accurately calculate the prevalence. Thus, downstream analyses have been performed comparing the percentage of positivity calculated for each location at each sampling time. The frequency of CoV detection varied widely depending on target colony and sampling occasions, ranging between 0 and 65% (see S8 Table in supporting informations). In particular, colony A turned out positive only in 2 out of 6 sampling occasions, with percentage of positivity of 4.5 and 16.7% (overall mean 5.3%). We found the opposite situation in colony B, where 4 out of 6 samplings were positive with a range of positivity spanning from 5 to 35% (overall mean 9.2%). Finally, colony C showed the highest frequency, with 5 out of 6 positive samplings and percentage of positivity for CoVs ranging from 15 to 65% (overall mean 31.7%). Regardless these differences, the Kruskal-Wallis test estimated no significant differences between colonies (p-value = 0.07038). Due to challenges in standardizing the collection in colony A, we evaluated the influence of the sampling month within colonies B and C only. Statistical analyses (K proportion test) highlighted that August yielded a statistically significant positivity

(p-value = 0.001) compared to the other events, while in colony C this association was observed (p-value <0.001) in the month of May. In colony C, the percentage of positivity decreased as July progresses and finally increased again towards the end of August (see S8 Table in supporting information and Fig 4A in the main text).

Sequencing of positive amplicons obtained from the three colonies of *P. kuhlii* investigated in this study revealed that most viruses belong to the species BTCoV_020, with a single detection of Pk-BtCoV in colony C in the month of June, when this variant seemed to temporarily replace BTCoV_020. Phylogenetic analyses showed low variability and no phylogenetic clustering based on sampling location or time (Fig 4B). Interestingly, Pk-BtCoV clustered altogether with strains detected in Europe from the same species *P. kuhlii,* with particular note from sequence MW089336, detected in 2018 in Italy from colony A (Fig 4B). However, further screening of this colony performed during this study failed to detect this variant in this location during the screening performed in 2022.

**Estimate of the bat CoVs' diversity in swine farms from North Eastern Italy.** All eight bat species featured in our bioacoustic surveys have been associated with at least one coronavirus whose sequence was deposited on GenBank (see S9 Table in supporting informations). Based on genetic and phylogenetic analysis of these published sequences, at least three different species of CoVs circulate in the majority of target bat species, often spread across their distribution range. The highest viral diversity observed in *E. serotinus* and *R. ferrumequinum*, where respectively five and six distinct coronaviruses were identified (Fig 5).

If we consider that bats circulating in Italian piggeries might be infected with viruses detected in the same species elsewhere, at least eight CoV species at this interface, including six *Alphacoronavirus* and two *Betacoronavirus.* Among alpha-CoVs, four have been officially classified by ICTV, namely *Alphacoronavirus pipistrelli* (Pk-BatCoV), belonging to the subfamily *Nyctacovirus*, *Alphacoronavirus ferrumequini* (BtRf-AlphaCoV), belonging to the subfamily *Decacovirus*, *Alphacoronavirus rhinolophi* (HKU2), belonging to the subfamily *Rhinacovirus*, and *Alphacoronavirus finnoniense* (btCoV_020), belonging to the subfamily *Pedacovirus*. In addition, target bat species are associated with the new lineage 1 (BatAlpha_NL1) recently identified in serotine bats as belonging to a new subfamily that is yet to be officially designated [68]. Finally, our analysis underlined the presence of a further cluster of alphaCoVs in the bat *E. serotinus* whose divergence support the classification as a novel species and subgenus (Fig 5). Unfortunately, no whole genomes were available to better characterize this cluster. Among betaCoVs, target bat species can be associated with *Betacoronavirus cameli* (MERS-CoV) and *Betacoronavirus pandemicum* (SARSr-CoV). Beside SARSr CoV that was found exclusively in members of the genus *Rhinolophus*, other CoV species were found to circulate in more than one host, including bats belonging to different genera and families. However, phylogenetic analyses support subspecies clustering upon the host genera in all cases, with the relevant exception of viruses shared between *P. kuhlii* and *H. savii* (Fig 5).

We attempted whole genome sequencing of viruses found in these two species in Italy, also exploiting positive archive sample available at Istituto Zooprofilattico Sperimentale delle Venezie. Unfortunately, samples from *Pipistrellus kuhlii* positive for Pk-BatCoV and btCoV_020, including feces from this study and organs from routine passive surveillance [58] showed low quality and quantity of RNA, and no MERS-CoV were found in this host species between 2018 and 2022. While we found MERS-CoV in two organs from *Hypsugo savii* from routine passive surveillance, their sequencing only produced partial genomes, whose *RdRp* have been deposited in Genbank and analysed within the framework of this study (GenBank accession numbers PV420065 and PV420066). On the other hand, we obtained and annotated the whole genome of BtCoV_020 from an *H. savii* using a metagenomics approach (Genbank accession number of consensus sequence PV420123, SRA accession number of raw PRJNA1234035). The virus followed the same structuring of *Pedacovirus* (see S3 Fig in supporting informations) and showed aminoacidic identity at the eight conserved domains of ORF1ab between 92.7% and 95.3% (mean 94.3%) with the closest relative of BtCoV_020. Phylogenetic analyses showed basic congruence over ORF1ab and structural proteins, with the Italian strain clustering within BtCoV_020 but not with PEDV together with two sequences detected in *Pipistrellus nathusii* from Russia (GenBank accession number: OQ230639 and OP919651) and a sequence detected in UK from *P. pipistrellus* (GenBank accession number: OQ401253) (see S4 Fig in supporting informations).

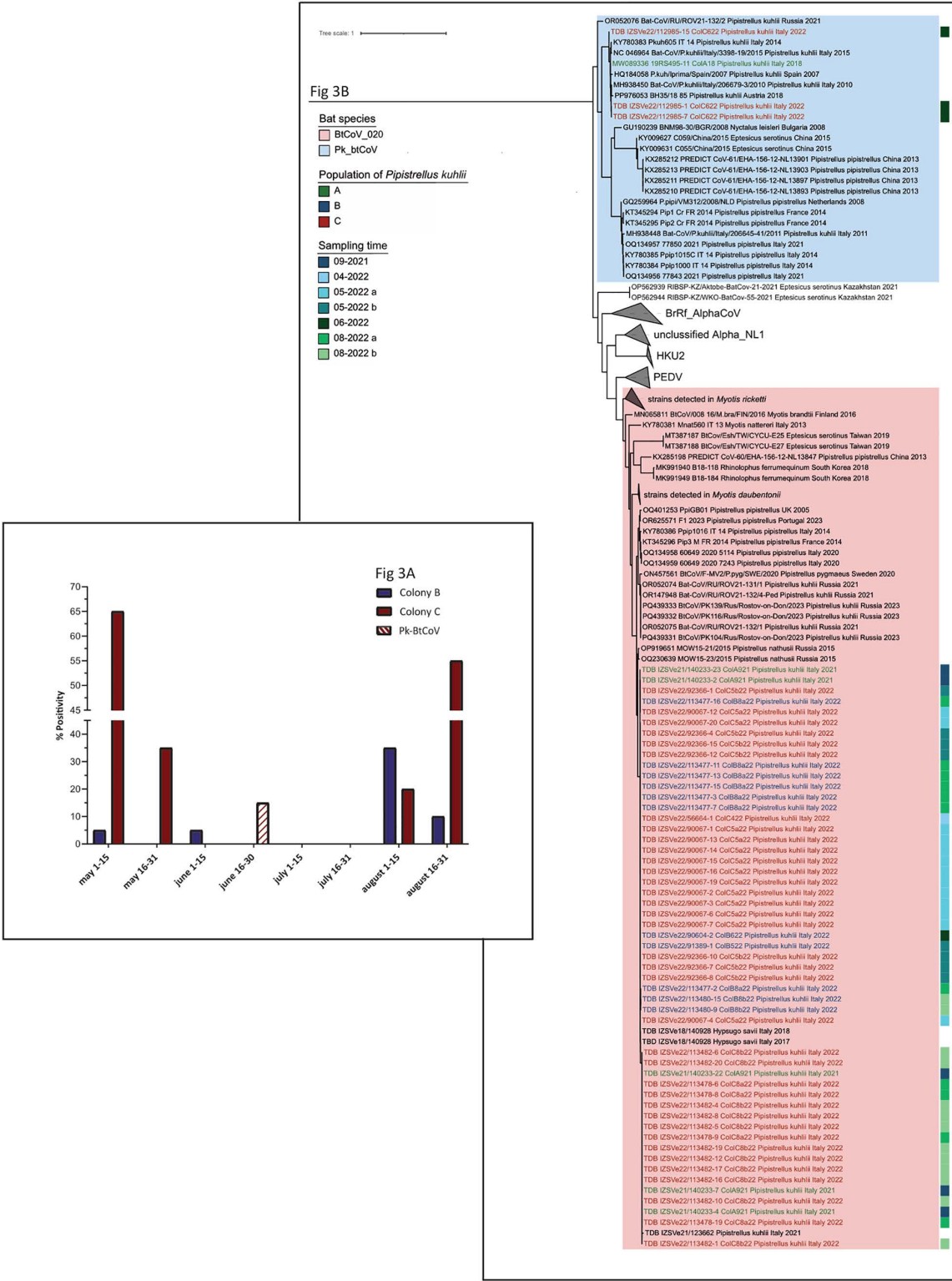

**Fig 4. Seasonal trends and phylogenetic tree of coronaviruses from active and passive surveillance. A: Seasonal trend for the shedding of coronaviruses determined in colonies B and C.** The graph is a summarized version of the S8 Table in supporting informations; colony B is identified in blue and colony C in red. The bar highlighted with the striped pattern in white and red is representative of the sampling event in which the cluster

of Pk-BatCoV was detected in colony C. The figure have been obtained using Graphpad PRISM. **B: ML Phylogenetic tree of Alphacoronaviruses found during active surveillance of Italian *P. kuhlii***. The tree is a pruned version of Fig 4 obtained using iTOL, with clusters of Pk-BatCoV in blue and BTCoV_020 in red. We coloured the sequences labels according to the sampling colonies and produced a colour strip ranging from green to blue to indicate different sampling occasions, as shown in the figure.

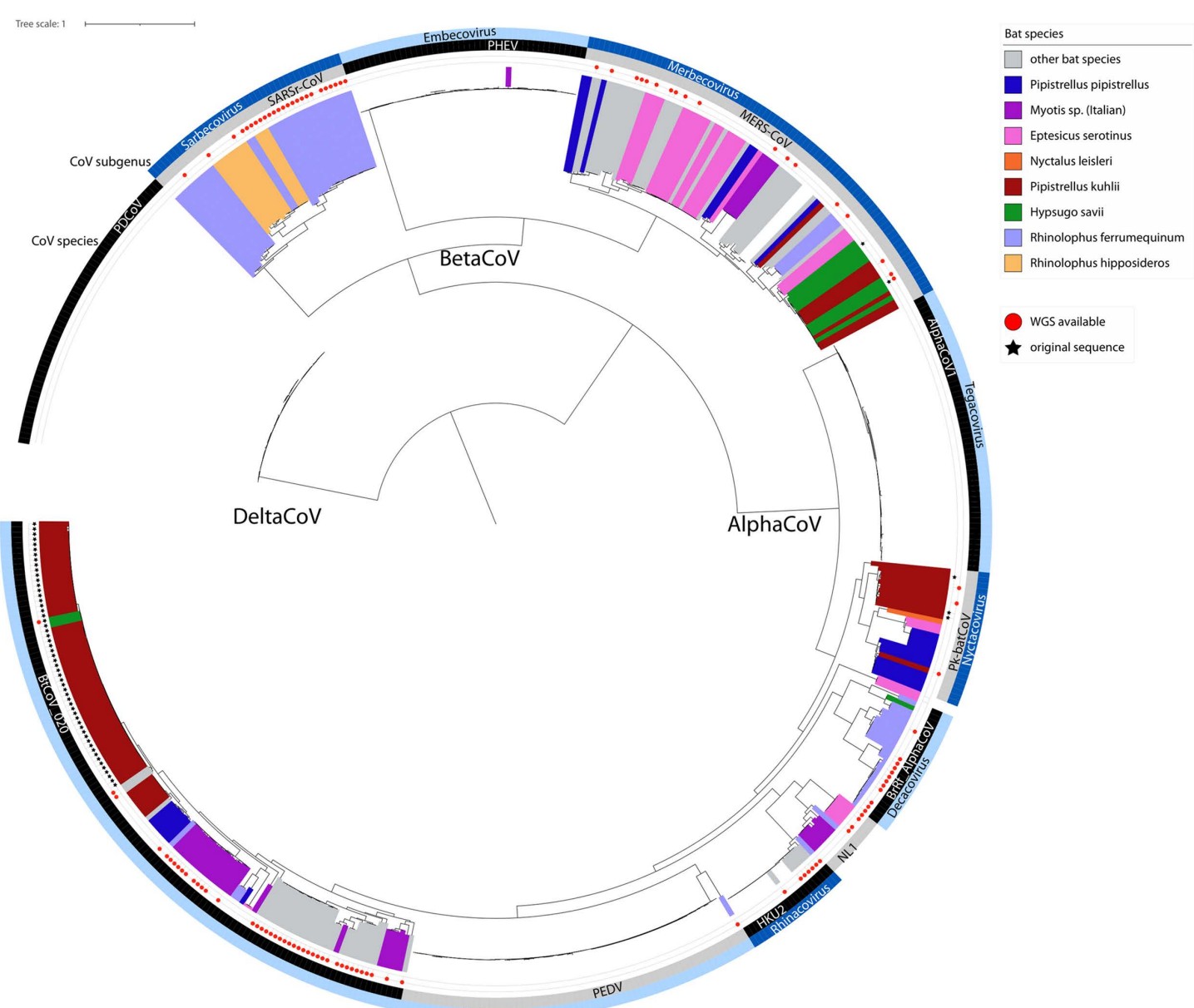

**Fig 5. ML Phylogenetic tree of coronaviruses potentially circulating in swine and bat species detected in Italian piggeries, including original sequences from this study and sequences retrieved from public repositories.** We coloured the sequences according to the host species, as detailed, and indicated i) with a star the sequences obtained through this study and ii) with a red dot the sequences for which WGS is available. Outer circles show the CoV species and subgenus.

## Discussion

In this study, we explored the possible spillover of CoVs at the interface between bats and pigs in intensive farming systems in North Eastern Italy through a holistic and multidisciplinary approach applying methodologies from ecology, virology and epidemiology.

Based on previous frameworks on pathogen spillover from wildlife, we wanted to evaluate i) the distribution and density of reservoir hosts, ii) the presence of structural barriers between reservoir and target host species and iii) the prevalence of infection and the shedding of target viruses [70].

### Distribution and density of reservoir hosts

The use of bioacoustics analyses allowed us to define both the frequency and the composition of the bat's communities in the pig farms of the area and the specific use of piggeries.

We recorded eight species among the 25 reported in the region [71]. This data is lower compared to previous reports which provide acoustic evidence for the circulation of at least 13 bat species in Italian piggeries [54]. While it should be noted that, compared to Leopardi et al. [54], in this study we were unable to discriminate between different species of the genus *Myotis,* here accounted as a single species, lower richness detected in our survey might be due to profound difference in the study design of the two works and to limitations of our sampling methodology caused by complication from the H5N1 and ASF epidemics. Indeed, the preliminary report targeted farms considered suitable for bats prior to the survey [54], compared to this study where piggeries were selected randomly, likely providing a more accurate picture and correcting for previous overestimation. This hypothesis is corroborated by the variation of richness within single farms analysed in our survey, ranging from a single species detected in farm n 7 to higher bat diversity of six species in farm n 8, that was similar to the one detected in the previous survey [54]. *P. kuhlii, P. pipistrellus,* and *H. savii* were the most common species in this agro environment, underlining the role of pig farms as valuable feeding and roosting sites for these species, in line with findings from cattle farming [72,73]. These three species have been undergoing a notable range expansion in recent years, frequently occupying various niches, especially in human-dominated habitats [74]. Occasionally, we also detected species from the genera *Myotis, Rhinolophus, Eptesicus*, and *Nyctalus*, similarly to what have been observed in similar environmental contexts elsewhere [54].

Overall, these data confirm previous evidence that some species are well-adapted to agro environments and can be find with high frequency and intensity in these settings [75,76] even though bat richness is often reduced by intensive agriculture and livestock farming [47,77]. However, evidence that richness is highly variable between study sites suggests that farms differ in their suitability for bats and that some can provide suitable ecological niches in otherwise hostile environments of intensive agriculture.

We performed ecological and statistical modelling to define the variables that explain differences in bat richness and activity detected in the bioacoustics survey. We hypothesized that such a difference could be either related to the surrounding habitat or to specific characteristics of farms that may provide suitable shelter, feeding and drinking grounds. For example, previous studies suggested that presence of crevices, holed bricks, gutters and other structure may create structurally complex islands that can be exploited by synanthropic bat species but also by some specialists that usually occupies peculiar niches, leading in return to a local increase in bat species richness and activity [78]. Our models showed that the habitat did not heavily affect either the activity of bats or their richness. This result can be due to the actual lack of variability in our system as determined by landscape analysis, which showed how sampled swine farms are mostly immersed in a human dominated context where natural habitats such as swamps and lacunar systems have been largely devolved to intensive agriculture and farming. In Italy, this situation is spread to most of the Po Valley, in Northern Italy that host the greatest density of intensive farms for most animal species, including swine, cows and poultry [79]. Thus, while our results might not fully reflect the whole reality of the country, we believe that they are likely representative of farms facing more severe consequences in case of disease introduction and at highest risks for disease spread in the country [80,81].

On the other hand, some structural features of the farms showed statistical influence in bat richness, bat frequency or both. For example, the presence of empty rooms consistently enhanced bat activity, both in general and specifically for *P. kuhlii.* We suggest that empty rooms might provide a safe space for bats that shelters from predators and simultaneously is attractive to insects particularly since is a common practice by farm workers to use these spaces to stock pig fed. This hypothesis is also corroborated by the fact that other features known to shift insect communities such as the number of pigs and the presence of bulb-lights also showed positive correlation with overall bat activity. This result is consistent with previous data that some species can exploit livestock and bulb-lights for feeding activities [20,69]. The scarce levels of feeding-buzzes recorded in this survey, peaking with a maximum of 5% in one farm of our acoustic sample (mean = 1%) and the lack of significance of this parameter for *P. kuhlii*, which should benefit the most of artificial illumination based on the literature [69,82], would seems in contrast with this hypothesis. Authors acknowledge that this finding might be confounded by different suitability for the species related with the intensity and the colour of lights across farms. Unfortunately, we could not account for these differences because the data was not collected during surveys. In addition, the scarcity of feeding buzzes recorded in our study may have also been affected by the position of the bat detector, usually installed to privilege flight corridors, as well as microphone orientation and the weather, turning out in the difficulty of catching feeding sounds, as very well documented by Staton & Poulton [83]. On this matter, a higher feeding activity was previously reported by Leopardi and colleagues [54] in a similar scenario. Therefore, we suggest that more punctual investigations are needed to ascertain the real impact of artificial lighting on bat foraging behaviour in these agro-environment, considering variables such as light intensity, spectrum, placement of detectors, and interactions with local insect dynamics. Albeit we did not find evidence for seasonal variability of acoustic data, the single sampling performed for each location affected our ability to fully capture the temporal variability in bat activity and species richness and represents another limitation of this study. While we could not overcome this issue due to restricted access after the introduction of ASF in the country [42], future studies should be performed with larger temporal coverage and sample size. Finally, the use of external batteries could allow enlarging the sampling window for each night, allowing a better capture of bat diversity according to different ecology of species.

## Presence of structural barriers between reservoir and target host species

Regardless the frequency and biodiversity of bats inhabiting piggeries, the likelihood of viral spillover from these animals relies on the occurrence of actual contacts between reservoir and target species, including direct interactions and exposure with droppings. In our case, we noted how windows are mostly opened at night to favour ventilation, with no grids or nets preventing bats or bat faeces to enter. In addition, bats can be found flying or roosting within feed storage rooms, providing chances for indirect transmission. In this situation, further ecological and virological data are needed to carefully assess routes for the exposure of pigs to bat viruses and design effective biosafety measures that could help controlling pathogen spillover even in farms characterized by high bat biodiversity and activity. This approach has been extensively implemented in Italy to control the introduction of highly pathogenic avian flu from wild birds to poultry, with various success [80,84]. In addition, reducing the proximity of fruit bats to pig farms by banning the presence of trees within the farm or its perimeter has been effective in breaking the pathway leading to the spillover of the Nipah virus in Malaysia in 1997 [85], with no further cases detected after the initial control of the disease trough swine culling.

## Prevalence of infection and shedding in the reservoir

Fortunately, even if we described the presence of a risky interface that potentially allows the spillover of bat pathogens to swine in Italy, no epidemics have been recorded in Europe. In addition, a parallel study on CoV diversity performed in the target piggeries provided no evidence for the circulation of bat coronaviruses (Festa et al., in preparation). However, we must acknowledge that sampling size of this work would have been effective in showing viruses circulating at a minimum prevalence of 5%, which could not reflect rare spillover events. On the other hand, no serological approaches could be

developed that could increase the sensitivity of such surveillance. The emergence of bat pathogens in swine populations have been documented elsewhere, causing severe consequences for both animal and public health. Other than the aforementioned Nipah virus, causing lethal cases in swine as well as in humans, the bat coronavirus HKU2 emerged in Chinese swine industry between late 2016 and 2019, causing a severe gastrointestinal disease called SADS [86]. While henipaviruses, up to date, have only been detected in fruit-bats of the genus *Pteropus*, CoVs have been widely described in different bat species worldwide, Italy included.

In this study, we aimed at evaluating the circulation of bat CoVs in bats circulating in swine farms. For this task, we focused on *Pipistrellus kuhlii* that was found to be the most spread, frequent and active species at the target interface in our territory. In addition, field investigation confirmed its roosting within piggeries, increasing chances for risky interactions. All colonies investigated for this work turned out positive for CoVs, showing the circulation of two distinct species, namely BtCoV_020 and Pk-BatCoV, both belonging to the genus *Alphacoronavirus*. On the other hand, we found no evidence for the circulation of the *Betacoronavirus* MERS-CoV that has been, however, already confirmed in Italy [87]. The percentage of positivity varied widely according to different sites (three in total) and sampling occasions. However, challenges in the capture and individual sampling of these animals hampered to define true prevalence over time. In two colonies (B and C) we could design and standardize an environmental sampling able to minimize the influence of environmental conditions while increasing the statistical quality of the data. In particular, cleaning the surface in the evening allowed us to collect only fresh faeces, while the collection of single droppings far from each other was used as a proxy for individual sampling. In these colonies, we were able to detect shedding of CoVs between May and early June and again in August, with two peaks of amplification that were particularly evident in colony C, which also showed the highest percentage of positivity across the whole study period. This data is consistent with the literature available for CoVs in other bat species [88]. In addition mathematical models obtained for the circulation of lyssaviruses in the greater mouse-eared bat (*Myotis myotis*), also suggested two peaks of virus amplification, one in spring upon the formation of the colony and one in late summer fuelled by the reproductive pulse [89] Longitudinal surveillance seems to support very low (often undetectable) circulation in summer between June and July, with the sole exception of 15% positivity in colony C in late June. Interestingly, this peak was associated with the emergence of Pk-BatCoV, while all other findings described the shedding dynamics of BtCoV_020. This result highlights how sequencing should always follow molecular detection of CoVs using broad-spectrum screening and suggests that different CoV species might differ in their ecology. The detection of different CoV species in the same population, albeit not concomitantly, warrants further attention because it may facilitate recombination.

As noted, the diversity of CoVs should be fully considered when assessing risks for pathogen spillover. After the first discovery of SARSr CoVs in bats from the genus *Rhinolophus,* the host-specific association of coronaviruses became more and more evident [37]. While further surveillance and more detailed taxonomic studies later revealed that several CoV species can infect more than one host and, conversely, several bat species can be infected by multiple ones, the preferential transmission of CoVs within single bat species still holds [68]. In turn, we hypothesized that swine are exposed to several bat CoVs depending on the local biodiversity of bats, increasing chances for one of these actually being able to infect them. By performing genetic and phylogenetic analyses using sequences associated with the bat species detected by bioacoustics analyses, we identifies eight CoV species of interest due to the presence of their preferential reservoir host, each one further divided in different strains evolved in different hosts. Of note, our phylogenetic data support high level of inter-species transmission of viruses between *P. kuhlii* and *H. savii,* which share the same variant of MERS-CoV and BtCoV_020 with no species-specific clustering. For these species, belonging to distinct genera classified among Vespertilionidae, similar ecological features, such as the degree of sociality, the roosting ecology and feeding habits likely creates environments for CoVs transmission [90]. While sharing of MERS-CoV between these synanthropic bats was previously observed [91], we described high circulation also of BtCoV_020, which had previously been mainly associated with the genus *Myotis* [68]. This virus belongs to the subgenus Pedacovirus together with PEDV, which is currently considered one of the major infectious threat for the pig industry due to its high pathogenicity and epidemic potential.

Recent recombination analyses showed major recombination events within this subgenus [68], suggesting a possible risk related with the circulation of different species at the interface between bats and pigs. In addition, the emergence of swine enteric coronavirus (SeCoV) in Europe highlighted the ability of Pedacoviruses to recombinate across genera, with this swine CoV being characterized by a backbone of the Embecovirus TGEV and the spike of PEDV [92].

## Conclusions

Our study investigated the spillover of viruses from bats to pigs using solid frameworks described elsewhere [70].

While we obtained informations that can be used for a preliminary risk assessment, further investigation are needed to strengthen the findings of this study. Particular emphasis should be placed on the inclusion of structurally diverse pig farms, located in different regions in Italy and on the development of a detailed, stratified protocol for active surveillance prolonged over several year to better catch seasonal trends. In addition, efforts should be placed in the development of serological assays able to investigate exposure of swine to bat CoVs. Similar approaches have been developed for high impact SARSr CoVs, while attempts for other CoV species with unknown zoonotic and epizootic potential have up to now proved unsuccessful.

Briefly, we detected relevant biodiversity and density of bats within piggeries, describing structural parameters that influence the suitability of different settings. We spotted potential breaches of biosecurity that likely fail to block the exposure of pigs to CoVs and other bat viruses that were not targeted in our survey. We also confirmed the circulation of CoVs throughout the season and described seasonal peaks in shedding. While our surveillance focused on *P. kuhlii* and was then able to detect only two CoV species that are typical of this bat, we used bat biodiversity detected using bioacoustics to infer the possible diversity of CoVs. This result is critical for risk assessment, to design further targeted surveillance and to eventually develop serological assays to investigate pigs. We suggest that risk assessment should be completed with more detailed investigation on routes for exposure in order to design interventions and biosafety measures able to mitigate likelihood of cross-species transmission while maintaining the ability of farms to provide shelter and food to bats in agriculture hostile environments, which is particularly important for endangered species that are highly impacted by the intensification of farming.

## Supporting information

**S1 Table. Descriptive summary of the variables used in the study with the relative metrics.**
(DOCX)

**S2 Table. Descriptive summary of variables included in the analyses.**
(DOCX)

**S3 Table. Description of the acoustic sample divided per farm.**
(DOCX)

**S4 Table. Description of the acoustic sample divided per farm and per species.**
(DOCX)

**S5 Table. Descriptive summary of the results of the landscape analysis, divided per farm.**
(DOCX)

**S6 Table. Results of the model averaging when allowing up to 2 predictors.**
(DOCX)

**S7 Table. Results of the model averaging when allowing up to 4 predictors.**
(DOCX)

**S8 Table. Summary table of virological analysis.**
(DOCX)

**S9 Table. Sequences used for phylogenetic analyses.**
(DOCX)

**S1 Fig. Ternary plots of predicted overall activity, *P. kuhlii* activity and richness as a function of land-use in each buffer.**
(DOCX)

**S2 Fig. Relationship between observed and predicted activity, *P. kuhlii* activity, richness, and each predictor variable in the dataset.**
(DOCX)

**S3 Fig. Annotated structure of the novel genome of BtCoV_020.**
(DOCX)

**S4 Fig. Phylogenetic trees showing the correlation of novel strains of Italian BtCoV_020 with reference sequences based on 8 ORF1ab structural proteins, ORF3, S, E, M and N proteins.**
(DOCX)

## Acknowledgments

Authors want to thank all farmers and local veterinarian that allowed us to perform fieldwork in the region. We thank Maria Serena Beato, Denis Vio and Marzia Mancin from IZSVe for logistical help in the selection of farms. Also, we acknowledge the work of Maria Varotto and Silvia Ormelli for the Sanger sequencing of coronaviruses. Finally, we are grateful to Francesca Ellero for her language revision of the manuscript.

## Author contributions

**Conceptualization:** Pamela Priori, Paola De Benedictis, Stefania Leopardi.

**Data curation:** Francesca Festa, Gianpiero Zamperin, Francesca Cosentino, Pierre Nouvellet, Stefania Leopardi.

**Formal analysis:** Francesca Festa, Pamela Priori, Giulia Chiarello, Elisa Palumbo, Gianpiero Zamperin, Francesca Cosentino, Pierre Nouvellet.

**Funding acquisition:** Paola De Benedictis, Stefania Leopardi.

**Investigation:** Francesca Festa, Pamela Priori, Dino Scaravelli, Pierre Nouvellet, Stefania Leopardi.

**Methodology:** Francesca Festa, Pamela Priori, Gianpiero Zamperin, Francesca Cosentino, Luigi Maiorano, Stefania Leopardi.

**Project administration:** Paola De Benedictis, Stefania Leopardi.

**Resources:** Stefania Leopardi.

**Software:** Francesca Festa, Gianpiero Zamperin, Luigi Maiorano, Pierre Nouvellet, Stefania Leopardi.

**Supervision:** Francesca Festa, Pamela Priori, Francesca Cosentino, Luigi Maiorano, Maria Luisa Menandro, Paola De Benedictis, Stefania Leopardi.

**Validation:** Francesca Festa.

**Visualization:** Francesca Festa, Giulia Chiarello, Elisa Palumbo, Francesca Cosentino, Stefania Leopardi.

**Writing – original draft:** Francesca Festa, Pierre Nouvellet.

**Writing – review & editing:** Francesca Festa, Pamela Priori, Francesca Cosentino, Luigi Maiorano, Maria Luisa Menandro, Dino Scaravelli, Paola De Benedictis, Pierre Nouvellet, Stefania Leopardi.

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
