## [Decision Letter · Decision Letter 0]

7 Aug 2025

Dear Dr. Festa,

Thank you for submitting your manuscript to PLOS ONE. After careful consideration, we feel that it has merit but does not fully meet PLOS ONE’s publication criteria as it currently stands. Therefore, we invite you to submit a revised version of the manuscript that addresses the points raised during the review process.

We look forward to receiving your revised manuscript.

Kind regards,

Gábor Kemenesi, Ph.D.

Academic Editor

PLOS ONE

Journal Requirements:

2. To comply with PLOS One submissions requirements, in your Methods section, please provide additional information regarding the experiments involving animals and ensure you have included details on (1) methods of sacrifice, (2) methods of anesthesia and/or analgesia, and (3) efforts to alleviate suffering.

6. Thank you for stating the following financial disclosure:

The present work was supported by the First International ICRAD call under grant agreement N◦ 862605, ID 95 ConVErgence.

7. We note that Figure 1 in your submission contain [map/satellite] images which may be copyrighted. All PLOS content is published under the Creative Commons Attribution License (CC BY 4.0), which means that the manuscript, images, and Supporting Information files will be freely available online, and any third party is permitted to access, download, copy, distribute, and use these materials in any way, even commercially, with proper attribution. For these reasons, we cannot publish previously copyrighted maps or satellite images created using proprietary data, such as Google software (Google Maps, Street View, and Earth). For more information, see our copyright guidelines: http://journals.plos.org/plosone/s/licenses-and-copyright.

8. Please remove all personal information, ensure that the data shared are in accordance with participant consent, and re-upload a fully anonymized data set.

Reviewers' comments:

Reviewer's Responses to Questions

**Comments to the Author**

1. Is the manuscript technically sound, and do the data support the conclusions?

Reviewer #1: Yes

Reviewer #2: Partly

2. Has the statistical analysis been performed appropriately and rigorously?

Reviewer #1: N/A

Reviewer #2: Yes

3. Have the authors made all data underlying the findings in their manuscript fully available?

Reviewer #1: Yes

Reviewer #2: Yes

4. Is the manuscript presented in an intelligible fashion and written in standard English?

Reviewer #1: Yes

Reviewer #2: Yes

Reviewer #1: Authors analysed the bat-swine interface, to highlight a potential transmission route of bat coronaviruses to swine, and some related risk factors. The increasing relevance of swine farming in the food chain and the risks for pathogens spillovers from wildlife to swine, support the need for such studies. Under this light, this study contributes with some interesting still little explored information to aware of potential risks to pig farms. This study follows a previous one (doi: 10.3390/v13010004), based on more limited data, improving the data and, therefore, it appears of interest. In some parts it tentatively tried to describe the potential spillover of bat coronaviruses to pigs, even if overall it does not include data on these viruses in pigs, despite the potentially described chances of interaction between species.

Overall, it appears well written and methodologically supported, even if some parts could benefit of synthesis for more clarity. I below added some additional comments and suggestions for the Authors.

Major comments:

- I suggest defining the names of bat species in italics.

- Line 94: To avoid unnecessary misinterpretations, I suggest clarifying if these two cited novel genera are taxonomically officially recognized or tentatively defined with these proposed genera.

- Lines 126-128: I suggest briefly including more details on the investigated pig farms, to be more descriptive with such a wide audience of readers, not limited to this specific geographical context.

- Line 131: Did Authors consider biosafety measures generically to influenza viruses or specifically only to H5N1 avian influenza virus? In any case, I suggest specifying also the virus name rather than the only viral genotype.

- Lines 177-178: Did Authors concretely observe the contamination with bat feces inside the pig housing areas or demonstrate/hypothesize a way of introduction? It doesn’t appear clear, as it was described.

- Lines 240-241: “frequently found”, as currently expressed, is not referred to this specific farming context, as well “more active” is not referred to, therefore I suggest clarifying this sentence. Moreover, is this reference referred to “all study sites”? Did this reference include information on the common observed bats close to pig farms in Italy, as reported in other studies cited below? This should be clarified, also to support the text at lines 284-285. Finally, was the “pooled faecal material” (cited at line 242) tested for the species of origin (as described at line 292)?

- Lines 371-372: As this sentence was expressed, it appears that the analysis was limited to those viruses that are documented to be responsible for interspecies transmission events. Is there any reference to support or is this a speculative hypothesis?

- Lines 616-617: this sentence is not clear enough: how can this data be beneficial for risk of the introduction of other (not bat-related) viruses in pig farms?

- Lines 666 and 720: Did this study evaluate CoVs in bats in close contact to swine farms or directly bat CoVs “in” swine farms? Did this study investigate spillovers of viruses from bats to pigs?

- Line 724: As this study is interesting, further efforts should be made probably also towards other types of pig farming, as rural and semi-extensive farming are widely diffuse, not only to Italy but also in other parts of the globe, with closer and less controlled human-pig contacts.

Minor comments:

- Line 91: I suggest using italics characters for “Nidovirales”.

- Line 92: I suggest adding a comma before “and Deltacoronavirus..”

- Line 267: what do “, 2020” and “for faeces” mean?

- Line 271: I suggest adding “titled” just after “paragraph”.

- Lines 288-289: I suggest replacing “in Istituto” with “at Istituto”, and to add “,Italy” just after “Venezie”.

- Line 381: Should “respectively” be included?

- Lines 529-532: I suggest using italics characters for the subfamily names.

- Line 557: I suggest including the related accession numbers.

- Lines 566 and 567: I suggest preferring “GenBank accession number” or “accession number” to “gb an”.

- Line 750: I suggest moving “Sanger” before “sequencing” or add “method” after “Sanger”.

Reviewer #2: The authors explored the possible spillover of CoVs at the interface between bats and pigs in intensive farming systems in Northeastern Italy through a holistic and multidisciplinary approach, applying methodologies from ecology, virology, and epidemiology. This is an interesting study that employs a sound multidisciplinary analysis methodology. However, the study has a significant sampling issue.

The dates on which the acoustic sampling was carried out are not specified, and apparently, only one sampling per farm was conducted during the period between April and October. It must be taken into account that variability in bat activity and diversity changes significantly across different times of the year and even between sampling events within the same period. In this regard, sampling must be carefully planned, and multiple surveys should be conducted throughout the April to October period. The part of the discussion that talks about the observed bat activity and diversity should be reviewed.

Furthermore, in order to reliably compare bat activity and species diversity across farms, sampling must be carried out within the same temporal window. It is not clear that the authors followed this approach.

Additionally, the recording period appears to be too short. It is important to consider that some species emerge later at night and that others use human structures for temporary roosting during nocturnal activity, potentially after midnight.

Nevertheless, the article provides valuable data, particularly regarding CoV circulation in P. kuhlii. I recommend that the authors explicitly acknowledge these sampling limitations and clearly state that the data presented are preliminary. Further studies will be necessary to complement and expand upon the findings reported in this work.

Specific Suggestions and Edits:

Page 2, Line 71: Please change water to humidity.

Page 2, Line 77: Please add the sentence:

“...can offer essential drinking opportunities and food resources for bats.”

Page 2, Lines 77–79: Replace:

“In fact, bats are particularly vulnerable to dehydration: due to their unique morphology and physiology, they happen to lose substantial amounts of water through their body surfaces”

with:

“In fact, bats are particularly vulnerable to dehydration due to their high body surface area in relation to body volume, and the presence of bare wing membranes, which facilitate transpiration.”

Page 5, Line 153: Please write sp. without italics.

Page 12, Line 379: Please replace bat sounds with bat calls.

Page 12, Line 388: Please write sp. without italics.

Page 20, Line 621: Please clarify the phrase “Why attractive to insects?” – more context or explanation is needed.

Page 23, Line 722: Please delete the word crucial – it may be too strong or unnecessary in this context.

**Do you want your identity to be public for this peer review?** For information about this choice, including consent withdrawal, please see our Privacy Policy

Reviewer #1: No

Reviewer #2: No

---

## [Author Response · Author response to Decision Letter 1]

8 Aug 2025

We thank editors and reviewers for their valuable comments and the appreciation of our work. We therefore submit a revised version where all comments have been addressed.

Journal Requirements:

New files were uploaded with corrected names. The style has been revised throughout the text

2. To comply with PLOS One submissions requirements, in your Methods section, please provide additional information regarding the experiments involving animals and ensure you have included details on (1) methods of sacrifice, (2) methods of anesthesia and/or analgesia, and (3) efforts to alleviate suffering.

Thank you for this comment. We now specified that no anesthesia has been performed during active surveillance, while applying best practices to avoid suffering and minimize disturbance. On the other hand, no animals have been sacrificed as all individuals included in passive surveillance are submitted as carcasses to the laboratory.

Done

As we stated how we deposited sequences in GenBank and SRA, we cannot fully understand this comment. This might be related with the temporary embargo on Genbank until the release of the paper that is common practice. However, we set final deadline for the embargo at September 1st 2026 after which sequences will be released anyway.

Thank you for this comment, we will check accordingly during the submission phases.

6. Thank you for stating the following financial disclosure:

The present work was supported by the First International ICRAD call under grant agreement N◦ 862605, ID 95 ConVErgence.

Done

7. We note that Figure 1 in your submission contain [map/satellite] images which may be copyrighted. All PLOS content is published under the Creative Commons Attribution License (CC BY 4.0), which means that the manuscript, images, and Supporting Information files will be freely available online, and any third party is permitted to access, download, copy, distribute, and use these materials in any way, even commercially, with proper attribution. For these reasons, we cannot publish previously copyrighted maps or satellite images created using proprietary data, such as Google software (Google Maps, Street View, and Earth). For more information, see our copyright guidelines: http://journals.plos.org/plosone/s/licenses-and-copyright.

Thank you for this comment. We actually did not access to copyrighted information while using the licenced software ArcGIS Pro that should hold all permission for their users. We now changed the figure caption and deleted the source within the figure. Would this be fine for you?

8. Please remove all personal information, ensure that the data shared are in accordance with participant consent, and re-upload a fully anonymized data set.

We agree that a fully anonymized dataset is key to guarantee privacy. The questionnaire responded to international regulation and has been submitted by farm owners. In addition, no personal data are shown in the study, where farms have been randomly labelled 1 to 14 so that no identification of exact sites is possible. We now specified this in the method section. While we double checked that no information have been mistakenly provided, we ask you to highlight any error from our side so that we can amend it.

Done

Reviewers' comments to the Author

Reviewer #1: Authors analysed the bat-swine interface, to highlight a potential transmission route of bat coronaviruses to swine, and some related risk factors. The increasing relevance of swine farming in the food chain and the risks for pathogens spillovers from wildlife to swine, support the need for such studies. Under this light, this study contributes with some interesting still little explored information to aware of potential risks to pig farms. This study follows a previous one (doi: 10.3390/v13010004), based on more limited data, improving the data and, therefore, it appears of interest.

Thank you for your appreciation

In some parts it tentatively tried to describe the potential spillover of bat coronaviruses to pigs, even if overall it does not include data on these viruses in pigs, despite the potentially described chances of interaction between species.

While we actually sampled swine, no bat viruses have been found and the work was devolved to a study of swine coronaviruses that is in preparation. We now acknowledge this study in both the results and the discussion but unfortunately we have no DOI yet to reference to this.

Overall, it appears well written and methodologically supported, even if some parts could benefit of synthesis for more clarity. I below added some additional comments and suggestions for the Authors.

Thank you for your appreciation of our work.

Major comments:

- I suggest defining the names of bat species in italics.

Done

- Line 94: To avoid unnecessary misinterpretations, I suggest clarifying if these two cited novel genera are taxonomically officially recognized or tentatively defined with these proposed genera.

Done

- Lines 126-128: I suggest briefly including more details on the investigated pig farms, to be more descriptive with such a wide audience of readers, not limited to this specific geographical context.

Thank you for this comment. We revised the text for more description about the pig farms considered.

- Line 131: Did Authors consider biosafety measures generically to influenza viruses or specifically only to H5N1 avian influenza virus? In any case, I suggest specifying also the virus name rather than the only viral genotype.

Thank you for your comments. The sentence has been revised for more clarity to highlight how during H5N1 (avian flu) epidemics access to swine farms was highly limited, while being completely blocked after the first case of ASF. This justified why we could not sample more, unfortunately.

- Lines 177-178: Did Authors concretely observe the contamination with bat feces inside the pig housing areas or demonstrate/hypothesize a way of introduction? It doesn’t appear clear, as it was described.

Thank you for this comment. We revised the text for more clarity. Indeed, we used open windows and lack of grids as risk factors allowing entrance of bats or faeces but we could not standardize the collection of information on the presence/absence of bats or faeces.

- Lines 240-241: “frequently found”, as currently expressed, is not referred to this specific farming context, as well “more active” is not referred to, therefore I suggest clarifying this sentence. Moreover, is this reference referred to “all study sites”? Did this reference include information on the common observed bats close to pig farms in Italy, as reported in other studies cited below? This should be clarified, also to support the text at lines 284-285. Finally, was the “pooled faecal material” (cited at line 242) tested for the species of origin (as described at line 292)?

Thank you for this comment. We revised the text for more clarity. These data refer to actual results coming from the ecological part of the study, even if was also supported by previous investigation referenced. We also specified the performance of genetic ID on pooled environmental samples as suggested

- Lines 371-372: As this sentence was expressed, it appears that the analysis was limited to those viruses that are documented to be responsible for interspecies transmission events. Is there any reference to support or is this a speculative hypothesis?

Thank you for noticing possible misunderstanding of this statement. Indeed, we consider that spillover is possible only for viruses that are actually present and we tried to express this concept, even if we realize that this is actually speculative as we might overlook unknown viruses. The sentence has been deleted to avoid misunderstanding.

- Lines 616-617: this sentence is not clear enough: how can this data be beneficial for risk of the introduction of other (not bat-related) viruses in pig farms?

Thank you for this comment. We revised the text for more clarity.

- Lines 666 and 720: Did this study evaluate CoVs in bats in close contact to swine farms or directly bat CoVs “in” swine farms? Did this study investigate spillovers of viruses from bats to pigs?

Thank you for this comment. We revised the text to distinguish the aim addressed in this study as to investigate the presence of bat coronaviruses in bats circulating at the interface with pigs in Italian pig farms. While we investigated pigs as well data are presented in another study on swine coronaviruses, while no bat covs were found

- Line 724: As this study is interesting, further efforts should be made probably also towards other types of pig farming, as rural and semi-extensive farming are widely diffuse, not only to Italy but also in other parts of the globe, with closer and less controlled human-pig contacts.

Thank you for this comment. We agree that enlarging our sampling is interesting and is worthy for future funding. This was not possible in ConVergence project as IZSVe only has territorial competence for animal diseases in the study area.

Minor comments:

- Line 91: I suggest using italics characters for “Nidovirales”.

Done

- Line 92: I suggest adding a comma before “and Deltacoronavirus..”

Done

- Line 267: what do “, 2020” and “for faeces” mean?

Thank you for noticing this. It was a typo that has been corrected

- Line 271: I suggest adding “titled” just after “paragraph”.

Done

- Lines 288-289: I suggest replacing “in Istituto” with “at Istituto”, and to add “,Italy” just after “Venezie”.

Done

- Line 381: Should “respectively” be included?

Added

- Lines 529-532: I suggest using italics characters for the subfamily names.

Done

- Line 557: I suggest including the related accession numbers.

Thank you for the suggestion, done.

- Lines 566 and 567: I suggest preferring “GenBank accession number” or “accession number” to “gb an”.

Replaced throughout the text

- Line 750: I suggest moving “Sanger” before “sequencing” or add “method” after “Sanger”.

Done

Reviewer #2: The authors explored the possible spillover of CoVs at the interface between bats and pigs in intensive farming systems in Northeastern Italy through a holistic and multidisciplinary approach, applying methodologies from ecology, virology, and epidemiology. This is an interesting study that employs a sound multidisciplinary analysis methodology. However, the study has a significant sampling issue.

The dates on which the acoustic sampling was carried out are not specified, and apparently, only one sampling per farm was conducted during the period between April and October. It must be taken into account that variability in bat activity and diversity changes significantly across different times of the year and even between sampling events within the same period. In this regard, sampling must be carefully planned, and multiple surveys should be conducted throughout the April to October period. The part of the discussion that talks about the observed bat activity and diversity should be reviewed. Furthermore, in order to reliably compare bat activity and species diversity across farms, sampling must be carried out within the same temporal window. It is not clear that the authors followed this approach.

Thank you for the comments. Despite the full agreement on the variability across sites and sampling periods concerning the phenology of the species, the rates of activity and the richness in the population recorded, we had to address several complications (H5N1 and ASF epidemics) that impede us to perform replicates of sampling session in the same farm throughout the sampling year and between years. This represents the main limitation of our study and we clearly stated in the revised text to underline the need, in further similar analysis, to perform sound bioacustic monitoring projects.

Additionally, the recording period appears to be too short. It is important to consider that some species emerge later at night and that others use human structures for temporary roosting during nocturnal activity, potentially after midnight.

Thank you for the comments, and we agree completely on this point. Moreover, we decided to set the instruments with this temporal window to maximize the performances in terms of sampling days as much as possible. Concerning the fact that the noise of the piggeries would in any case trigger the recorders to switch on, we minimize the battery usage to the restricted time frame of sunset-midnight. Future direction will be of considering external batteries to be linked to acoustic devices in order to fully perform a systematic and sound acoustic monitoring. This issue has been addressed in the revised text.

Nevertheless, the article provides valuable data, particularly regarding CoV circulation in P. kuhlii. I recommend that the authors explicitly acknowledge these sampling limitations and clearly state that the data presented are preliminary. Further studies will be necessary to complement and expand upon the findings reported in this work.

Thank you for these general comments, revision have been addressed in the text.

Specific Suggestions and Edits:

Page 2, Line 71: Please change water to humidity.

Thank you for this comment, although we believe that humidity is not the word we wanted to insert as we referred to the availability of feeding grounds, drinking sites and roosts. We changed in the text accordingly.

Page 2, Line 77: Please add the sentence:

“...can offer essential drinking

---

## [Decision Letter · Decision Letter 1]

27 Aug 2025

A multi-disciplinary approach to identify spillover interfaces of bat coronaviruses to pig farms in Italy

PONE-D-25-37477R1

Dear Dr. Festa,

We’re pleased to inform you that your manuscript has been judged scientifically suitable for publication and will be formally accepted for publication once it meets all outstanding technical requirements.

Kind regards,

Gábor Kemenesi, Ph.D.

Academic Editor

PLOS ONE

Additional Editor Comments (optional):

Reviewers' comments:

Reviewer's Responses to Questions

**Comments to the Author**

Reviewer #1: All comments have been addressed

Reviewer #2: All comments have been addressed

2. Is the manuscript technically sound, and do the data support the conclusions?

Reviewer #1: Yes

Reviewer #2: Yes

3. Has the statistical analysis been performed appropriately and rigorously?

Reviewer #1: Yes

Reviewer #2: Yes

4. Have the authors made all data underlying the findings in their manuscript fully available?

Reviewer #1: Yes

Reviewer #2: Yes

5. Is the manuscript presented in an intelligible fashion and written in standard English?

Reviewer #1: Yes

Reviewer #2: (No Response)

Reviewer #1: The authors have reviewed their manuscript and provided a detailed response letter. All changes made have improved the manuscript overall and can be considered satisfactory. I have no further comments to make to the authors.

Based on the above considerations, this manuscript is eligible for full and positive evaluation by the Editors.

Reviewer #2: Lines 635–641. The lower bat species diversity observed on the farms in this study is compared to the higher diversity reported in a previous study, in which farms were selected based on their suitability for bat presence. Bat diversity and activity at a given site can vary significantly from one day to another, and even more so across different times of the year. However, although the authors acknowledge the limitations of the sampling methodology employed due to several complications (H5N1 and ASF epidemics) that prevent them from performing replicate sampling sessions on the same farm, it is not possible to draw reliable conclusions regarding the causes of the observed lower species diversity. I recommend deleting the sentences 'lower richness detected in our survey… in the previous survey [55]' and instead writing: 'The lower richness detected in our survey might be due to limitations of our sampling methodology caused by complications from the H5N1 and ASF epidèmics.

I recommend that, in future studies, the authors revise their acoustic sampling methodology. Recordings should be conducted over a longer duration during the night, ideally for 4 or 5 consecutive nights. Additionally, sampling should be performed periodically—ideally at least once a month—throughout the bat activity period (spring to autumn).

**Do you want your identity to be public for this peer review?** For information about this choice, including consent withdrawal, please see our Privacy Policy

Reviewer #1: **Yes: ** Francesco Mira

Reviewer #2: No

---

## [Editor Report · Acceptance letter]

PONE-D-25-37477R1

PLOS ONE

Dear Dr. Festa,

I'm pleased to inform you that your manuscript has been deemed suitable for publication in PLOS ONE. Congratulations! Your manuscript is now being handed over to our production team.

Kind regards,

on behalf of

Dr. Gábor Kemenesi

Academic Editor

PLOS ONE